# Essential Oils from Côa Valley Lamiaceae Species: Cytotoxicity and Antiproliferative Effect on Glioblastoma Cells

**DOI:** 10.3390/pharmaceutics15020341

**Published:** 2023-01-19

**Authors:** Mário Pedro Marques, Beatriz Guapo Neves, Carla Varela, Mónica Zuzarte, Ana Cristina Gonçalves, Maria Inês Dias, Joana S. Amaral, Lillian Barros, Mariana Magalhães, Célia Cabral

**Affiliations:** 1Clinic Academic Center of Coimbra (CACC), Coimbra Institute for Clinical and Biomedical Research (iCBR), Faculty of Medicine, University of Coimbra, 3000-548 Coimbra, Portugal; 2Center for Innovative Biomedicine and Biotechnology (CIBB), University of Coimbra, 3000-548 Coimbra, Portugal; 3Chemical Process Engineering and Forest Products Research Centre (CIEPQPF), Faculty of Medicine, University of Coimbra, 3000-548 Coimbra, Portugal; 4Laboratory of Oncobiology and Hematology, University Clinic of Hematology and Applied Molecular Biology, Faculty of Medicine, University of Coimbra, 3000-548 Coimbra, Portugal; 5Group of Environment Genetics and Oncobiology (CIMAGO), Coimbra Institute for Clinical and Biomedical Research (iCBR), Faculty of Medicine, University of Coimbra, 3000-548 Coimbra, Portugal; 6Mountain Research Centre (CIMO), Polytechnic Institute of Bragança (IPB), Campus Santa Apolónia, 5300-253 Bragança, Portugal; 7Associate Laboratory for Sustainability and Technology in Mountains Regions (SusTEC), Polytechnic Institute of Bragança (IPB), Campus de Santa Apolónia, 5300-253 Bragança, Portugal; 8PhD Programme in Experimental Biology and Biomedicine, Institute for Interdisciplinary Research (IIIUC), University of Coimbra, Casa Costa Alemão, 3030-789 Coimbra, Portugal; 9CNC—Center for Neuroscience and Cell Biology, University of Coimbra, 3004-504 Coimbra, Portugal; 10Centre for Functional Ecology, Department of Life Sciences, University of Coimbra, 3000-456 Coimbra, Portugal

**Keywords:** essential oils, *Lavandula pedunculata*, *Mentha cervina*, *Thymus mastichina* subsp. *mastichina*, Côa Valley, cytotoxicity, glioblastoma

## Abstract

*Lavandula pedunculata* (Mill.) Cav., *Mentha cervina* L. and *Thymus mastichina* (L.) L. subsp. *mastichina* are representative species of the Côa Valley’s flora, a Portuguese UNESCO World Heritage Site. *L. pedunculata* and *T. mastichina* are traditionally used to preserve olives and to aromatize bonfires on Saint John’s Eve, while *M. cervina* is mainly used as a spice for river fish dishes. Despite their traditional uses, these aromatic plants are still undervalued, and literature regarding their bioactivity, especially anticancer, is scarce. In this work, the morphology of secretory structures was assessed by scanning electron microscopy (SEM), and the composition of essential oils (EOs) was characterized by gas chromatography-mass spectrometry (GC-MS). The study proceeded with cytotoxic evaluation of EOs in tumor and non-tumor cells with the cell death mechanism explored in glioblastoma (GB) cells. *L. pedunculata* EO presented the most pronounced cytotoxic/antiproliferative activity against tumor cells, with moderate cytotoxicity against non-tumor cells. Whereas, *M. cervina* EO exhibited a slightly lower cytotoxic effect against tumor cells and did not affect the viability of non-tumor cells. Meanwhile, *T. mastichina* EO did not induce a strong cytotoxic effect against GB cells. *L. pedunculata* and *M. cervina* EOs lead to cell death by inducing apoptosis in a dose-dependent manner. The present study suggests that *L. pedunculata* and *M. cervina* EOs have a strong cytotoxic and antiproliferative potential to be further studied as efficient antitumor agents.

## 1. Introduction

Glioblastoma (GB), the most common primary glioma of the central nervous system (CNS), is characterized by a highly heterogeneous and diffuse invading nature. According to the World Health Organization (WHO), this adult-type diffuse glioma is classified as a grade IV tumor (highly malignancy level) [1]. Despite the advances in the clinic to improve patient survival, the overall average survival rate for GB is only 15–21 months after diagnosis, with just a 5% probability of survival in the 5 years post-diagnosis [2,3,4]. These numbers are not surprising, being explained by the late diagnosis and poor prognosis of GB. Thus, to hinder this situation, the current first-line treatment available for GB includes maximal surgical resection followed by concomitant cycles of chemoradiotherapy and adjuvant chemotherapy (temozolomide (TMZ)) [3,4,5,6,7]. Although, these treatments have several limitations, such as incomplete surgical resection or development of multidrug resistance (MDR), that provoke patient relapse [8]. Hence, novel, efficient, and site-specific therapeutic approaches are urgently required, in which plant-based phytoproducts can be considered an interesting and appealing solution.

Since ancient times and across all populations, the therapeutic value of plants was validated by their wide use in traditional medicine to treat various diseases, including cancer [8,9]. The well-known therapeutic significance attributed to plants usually resides in the secondary metabolites they produce. These metabolites represent an important source of bioactive molecules in the process of discovery and development of new pharmacologically active compounds, particularly for cancer therapy [3,10,11]. In fact, several chemotherapy agents currently in clinical use derived from plants or were inspired in the chemical structure of plant-extracted compounds [12,13].

Essential oils (EOs), obtained from aromatic plant species, are examples of complex mixtures of volatile secondary metabolites with huge therapeutic potential [9]. They are produced and stored in specialized structures, internal or external, and are characterized by the presence of small molecular weight oxygenated compounds and hydrocarbons, mainly monoterpenes and sesquiterpenes. These compounds are well-known for their therapeutic value, including antitumor activity [10,14,15]. Hereby, some studies have explored the antitumor potential of EOs and their isolated compounds, either as chemotherapeutic agents or as adjuvants to conventional therapy, against cancer cells, uncovering their mechanism of action (e.g., inducing cell cycle arrest, apoptosis, and DNA repair) [10,16,17,18,19,20,21,22].

Lamiaceae is the sixth largest flowering plant family in the world, with more than 7000 species and 250 genera, showing a worldwide distribution and being recognized for several EO-bearing plants [2,23]. Moreover, the antitumor properties of EOs from plant species of the Lamiaceae family are widely recognized and reported [10,23,24].

The Côa Valley, in the Northeast of Portugal has been recognized as UNESCO World Heritage Site since 1998. Despite its paleolithic value, its natural resources should also be valorized and preserved. From this perspective, the following three Lamiaceae species: *Lavandula pedunculata* (Mill.) Cav. (*L. pedunculata*), *Mentha cervina* L. (*M. cervina*) and *Thymus mastichina* L. (L.) subsp. *mastichina* (*T. mastichina*) stand out given their prevalence in the territory, ethnobotanical uses and cultural value, which makes their medicinal traditional uses worthy of validation. *L. pedunculata* is an aromatic shrub widely spread throughout the west Mediterranean region, namely in the Iberian Peninsula, North Africa, and Turkey. In Portugal, it is mostly used in religious ceremonies to aromatize bonfires on Saint John’s Eve, as well for ornamental purposes. In traditional medicine, *L. pedunculata* has been used as a therapeutic agent with antiseptic action for cleaning wounds, in infusions for internal and external applications, and is recommended for the respiratory and digestive systems [25,26,27]. *M. cervina*, commonly known as Hart’s pennyroyal, is an aromatic herb that grows on the edges of flooded areas, riverbanks, and wet places. However, because of excessive harvesting and habitat destruction, natural populations have been disappearing. Its traditional use is mostly associated with food seasoning, namely river fish dishes, besides this, it is used for the prevention of gastric disorders and inflammations of respiratory origin [28,29,30]. *T. mastichina* is an Iberian endemic thyme species also used as a food spice [31], and with potential applications in the perfume and cosmetic industry [32]. It is also used in several health conditions due to its antiseptic, digestive, antirheumatic, and antitussive effects [32,33,34,35,36]. The pharmacological validation of this thyme has been focused on several biological activities, particularly anti-inflammatory [37], antioxidant [38], anti-carcinogenic [38] and antifungal [39]. 

Despite the great potential of these species as valuable sources of bioactive molecules, the literature survey revealed limited work regarding their EOs, namely their antitumor potential. Therefore, in order to overcome this liability, their chemical composition and the morphology of the secretory structures was fully addressed. Following this, the EOs from these three species were tested as potential cytotoxic agents using a robust panel of cell models of GB. Their tumor-specific cytotoxic profile was also assessed in a non-tumor cell line. The EOs with the most pronounced cytotoxic effects were further evaluated regarding their cell death profile and impact on cell cycle regulation.

## 2. Materials and Methods

### 2.1. Plant Material

Plant samples of *L. pedunculata* (Figure 1A), *M. cervina* (Figure 1C) and *T. mastichina* (Figure 1E) during their flowering stage, in April 2022, July 2021, and May 2021, respectively. The three studied plants were harvested in river Côa Valley (Portugal). Specifically, *L. pedunculata* was collected next to the Museum and Côa Parque Foundation (41.080504, −7.110663), *M. cervina* in Côa riverbanks of the paleolithic rock-art site of Penascosa (41.005644, −7.105894), while *T. mastichina* was collected near to the village of Muxagata (41.028215, −7.146840). Voucher specimens were deposited at the Herbarium of the University of Aveiro (AVE), under the numbers AVE156 for *L. pedunculata*, AVE425 for *M. cervina* and AVE221 for *T. mastichina*. After harvesting, the plants were kept in the dark, at room temperature (25 °C), until EOs extraction. Fresh plant samples were in turn used for scanning electron microscopy (SEM) analysis.

### 2.2. Essential Oils Isolation

EOs were extracted from the aerial parts in the flowering stage of the three studied species, by hydrodistillation for 3 h using a Clevenger-type apparatus, according to the protocol described in the European Pharmacopoeia [40]. The isolated EOs were stored at 4 °C in appropriate dark glass vials and kept in the dark.

### 2.3. Gas Chromatography-Mass Spectrometry (GC-MS)

The essential oils were analyzed by gas chromatography coupled to mass spectrometry detection (GC-MS) using a GC-2010 Plus (Shimadzu, Kyoto, Japan) system equipped with a AOC-20iPlus automatic injector (Shimadzu, Kyoto, Japan) and a SH-RXi-5 ms column (30 m × 0.25 mm × 0.25 μm; Shimadzu, Kyoto, Japan, USA). The conditions used were the described by Spréa et al. [41], as follows: the initial oven temperature was set at 40 °C for 4 min, then raised at 3 °C/min to 175 °C, then raised at 15 °C/min to 300 °C, and held for 7 min; the temperatures of the injector, transfer line and ion source were 260 °C, 280 °C and 220 °C, respectively. The ionization energy was 70 eV and a scan range of 35–500 u with a scan time of 0.3 s was used. A volume of 1 μL of each sample diluted 1:100 in hexane was injected with a split ratio of 1:40, with helium being used as the carrier gas, adjusted to a linear velocity of 30 cm/s. The compounds were identified by comparison of spectra with NIST17 mass spectral library and by determining the linear retention index (LRI) based on the retention times obtained for a mixture of n-alkanes (C8–C40, ref. 40147-U, Supelco) analyzed under identical conditions. The calculated LRI was compared with previously published data [10]. Compounds were quantified as a relative percentage of total volatiles using the relative area values directly obtained from peak total ion current (TIC). Analyses were performed in triplicate.

### 2.4. Scanning Electron Microscopy (SEM)

For SEM analysis, leaves of *L. pedunculata*, *M. cervina* and *T. mastichina* were segmented into small pieces and placed on adhesive carbon stickers (12 mm, Agar Scientific, Stansted, UK) above metallic stubs. Then, without further preparations, they were observed using a variable pressure scanning electron microscope (Flex SEM 1000, HITACHI, Raleigh, NA, USA) in freeze conditions (−20 °C) and at an accelerating voltage of 10.0 kV, as stated in [42].

### 2.5. Cell Culture

U87 (brain-like glioblastoma), A172 (glioblastoma), and H4 (neuroglioma) cell lines were generously supplied by Professor Carla Vitorino (Faculty of Pharmacy, University of Coimbra, Coimbra, Portugal). U118 (astrocytoma/glioblastoma) and U373 (astrocytoma/glioblastoma) cell lines were kindly gifted by Professor Maria Conceição Pedroso de Lima (Faculty of Sciences and Technology, Center for Neuroscience and Cell Biology, University of Coimbra, Coimbra, Portugal). HEK-293 (immortalized human embryonic kidney fibroblasts) was kindly provided by Professor Paulo Oliveira (Institute for Interdisciplinary Research, Center for Neuroscience and Cell Biology, University of Coimbra, Coimbra, Portugal). Cells were cultivated in Dulbecco’s Modified Eagle’s Medium-high glucose (DMEM-HG) (Biowest, Nuaille’, France) supplemented with 10% (*v*/*v*) of heat-inactivated fetal bovine serum (FBS) (Sigma, St. Louis, MO, USA), and 1% (*v*/*v*) of penicillin-streptomycin (Sigma, St. Louis, MO, USA). Cells were maintained in a humidified atmosphere at 37 °C and 5% CO_2_. Throughout the experiments, cells were used at 80% confluence, and monitored by microscope observation to detect any morphological changes.

### 2.6. Cell Viability

U87, A172, H4, U118, U373, and HEK-293 cell lines were seeded in 96-well plates (1 × 10^4^ cells/well) 24 h before treatment. Each cell line was treated with a range of concentrations of each EO diluted in medium and with the vehicle control (DMSO), and further incubated for 24, 48 and 72 h. To assess cell viability, a modified Alamar blue assay was used [43]. Briefly, a solution of DMEM-HG medium with 10% (*v*/*v*) of resazurin salt dye stock solution (Sigma. St. Louis, MO, United States) at 0.1 mg/mL concentration was prepared, and subsequently added to each well after 24, 48, and 72 h. After 4 h of incubation at 37 °C and 5% CO_2_, the absorbance of the plate was measured at 570 and 600 nm in a BioTek reader (BioTek Instruments, Inc., Winooski, VT, USA). Absorbance results were obtained by the Gen5 program. The cell viability was calculated according to the following equation:$$\mathrm{Cell}\;\mathrm{viability}\;\left(\%\right)=\frac{A_{570}-A_{600}\;\mathrm{of}\;\mathrm{treated}\;\mathrm{cells}}{A_{570}-A_{600}\;\mathrm{of}\;\mathrm{control}\;\mathrm{cells}}$$

Furthermore, half-maximal inhibitory concentration (IC_50_) values were determined using GraphPad Prism Software v.7.04 (GraphPad Software Inc., San Diego, CA, USA) [11].

### 2.7. Cell Death Assay

The evaluation of cell death was accomplished using the Annexin V (AV)/propidium iodide (PI) assay by flow cytometry as previously described in [44]. Briefly, cells were treated with two different concentrations (0.7 or 0.8 µL/mL) of either *M. cervina* EO or *L. pedunculata* EO and further incubated for 24 h. After incubation, cells were stained with AV-APC and PI according to the manufacturer’s protocol (Biolegend, San Diego, CA, United States). Cells were resuspended in 100 µL of binding buffer and further incubated with 5 µL of AV-APC solution and 2 µL of PI solution. Afterwards, cells were diluted in 400 µL of binding buffer. For data acquisition and treatment, as described by [45], a six-parameter, four-color FACS Calibur flow cytometer (Becton Dickinson, San Jose, CA) was used, and at least 10,000 events were collected using CellQuest software (Becton Dickinson, San Jose, CA). The results were analyzed with the Paint-a-Gate software and were expressed in percentages (%).

### 2.8. Cell Cycle Progression Assay

Analysis of cell cycle progression was assessed by flow cytometry using a PI/RNase solution (Immunostep, Salamanca, Spain), according to the manufacturer’s protocol. Briefly, cells were treated with two different concentrations (0.7 or 0.8 µL/mL) of either *M. cervina* EO or *L. pedunculata* EO and further incubated for 24 h. After incubation, cells were detached and fixed in 70% ethanol for 1 h at 4 °C. Then, cells were stained with 500 µL of PI/RNase solution. Results were acquired using CellQuest software and analyzed to calculate the percentage of the cell population in subG0/G1, G0/G1, S, and G2/M cell cycle phases.

### 2.9. Statistical Analysis

At least three independent biological experiments were performed, and results were expressed as mean ± SD. Statistical significance was determined by 1-way ANOVA with Dunnett post hoc multiple comparison test. A *p*-value of *p* < 0.05 was considered significant. The statistical analysis was performed using GraphPad Prism v7.04 (GraphPad Software, San Diego, CA, USA).

## 3. Results and Discussion

### 3.1. Essential Oils Composition

According to the performed chemical characterization, *L. pedunculata* EO (Table 1) was mainly characterized by a high percentage of oxygen-containing monoterpenes (73%), followed by monoterpene hydrocarbons (14.7%). The major identified compounds were camphor (39%), 1,8-cineole (10.9%) and α-pinene (6.9%). These results are in accordance with those formerly described. That is, Costa et al. [46] reported that the EO from *Lavandula pedunculata* subsp. *lusitanica* (Chaytor) Franco has camphor (40.6%) as a major constituent. Moreover, Zuzarte et al. [47] were paramount to elucidate the chemical polymorphism of Portuguese populations of *L. pedunculata*, highlighting the existence of two well-defined chemotypes regarding quantitative differences of 1,8-cineole and fenchone. In addition, the 1,8-cineole group was further subdivided into two other subtypes according to the percentages of camphor [47]. 

Given the narrow distribution of wild populations of *M. cervina*, there are only a few comprehensive reports regarding its EO chemical composition, as well as its pharmacological and bioactive properties [29,30,48,49,50]. According to our GC-MS analysis (Table 1), the EO of Hart’s Pennyroyal is fundamentally rich in pulegone (60%), followed by germacrene D (12%). This is in full agreement with previously reported data regarding pulegone percentage [29,30,48,49,50]. Pulegone-rich EOs usually present high toxicity, which makes them not so suitable for food industry or aromatherapy [29,50]. However, this does not discourage their interest for antitumor approaches.

Regarding the composition of *T. mastichina* EO (Table 1), oxygenated monoterpenes (74.7%) are the main group, followed by monoterpene hydrocarbons (19%). This EO has 1,8-cineole as main compound with 57%, followed by α-terpineol (7.2%) and fenchone (5.36%). Interestingly, in contrast to other thymes widely known for their intraspecific chemical variability [51], chemical characterization on *T. mastichina* EOs has demonstrated that the oxygenated monoterpene 1,8-cineole is the major compound [52,53,54,55,56].

### 3.2. Secretory Structures Morphology

EOs are generated and stored in specialized secretory structures each with different morphologies, functions, and distributions [10,25]. In this report, the abaxial leaf surface of the three Lamiaceae species under study was submitted to an analysis by SEM (Figure 1).

Firstly, regarding the acquired SEM micrographs of *L. pedunculata*, the presence of large glandular peltate and smaller glandular capitate trichomes, as well as a significant number of non-glandular trichomes, was observed (Figure 1B). These large multicellular peltate trichomes are constituted by several secretory cells arranged in a single circle. Interestingly, in other Lamiaceae plant species, a higher number of secretory cells in the peltate trichomes are arranged in two concentric circles, as reported in some *Origanum* species [59]. Furthermore, it is worth mentioning that Zuzarte et al. [25] corroborate our observations, and that the ramified mixed trichomes found in *L. pedunculata* have not yet been described in other species [25].

Regarding the abaxial leaf surface of *M. cervina*, non-glandular trichomes and glandular trichomes were found (Figure 1D). On the other hand, large multicellular peltate trichomes formed by up to 13 secretory cells stand out from other smaller and globose trichomes that were also depicted. Similar observations were previously reported, but other trichome types were also observed when analyzing different organs of this same mint species [29]. 

On the other hand, to the best of our knowledge, this is the first insight into the trichomes of *T. mastichina*. In the present study, SEM micrographs obtained from *T. mastichina* leaves revealed numerous non-glandular trichomes with a warty surface and an acute apex, covering typically and large multicellular peltate glandular trichomes (Figure 1F). However, SEM analysis of *Thymus lykae* depicted both glandular peltate and capitate trichomes, as well as short and elongated unbranched non-glandular trichomes [59]. Thereby, we suggest that a more detailed anatomical and ultrastructural analysis may assist in additional findings regarding the secretory structures of the Iberian endemic *T. mastichina*.

### 3.3. Evaluation of the Cytotoxic Effect of Essential Oils

To establish the cytotoxic/antiproliferative profile of EOs from *L. pedunculata*, *M. cervina*, and *T. mastichina*, their effect was assessed on cell viability using Alamar blue assay. We used a panel of five glioblastoma cell lines (U87, A172, H4, U118, and U373 cells) and a non-tumor cell line (HEK-293 cells) that were treated either with different concentrations (1.0, 0.9, 0.8, 0.7, and 0.6 µL/mL) of each EO or with DMSO (vehicle control). 

Regarding *L. pedunculata* EO, despite its evident pronounced antiproliferative/cytotoxic effect against tumor cells (Figure 2), the probable tumor-specific mechanism that is observed for *M. cervina* EO (Figure 3) was not verified here. Moreover, analysis of the results showed that *L. pedunculata* EO, at the concentrations of 0.9 and 1.0 µL/mL, induced a significant decrease on the viability of HEK-293 cells (Figure 2).

Thus, as expected, after 24 h treatment, all the tested concentrations of *L. pedunculata* EO had a strong impact on the viability of tumor cells, inducing a significant decrease in the metabolic activity of GB cells (* *p* < 0.05; ** *p* < 0.01; *** *p* < 0.001; **** *p* < 0.0001), being more visible for U118 cells (decreasing cell viability by 80–100%) (**** *p* < 0.0001). Nevertheless, after 48 h incubation, a slight increase of A172, U373, and U118 viability was observed, while for U87 and H4 cells the effect remained constant. Finally, after 72 h, the metabolic activity of GB cells did not undergo significant changes compared to the previous time-point, apart from the U87 cell line, where it increased for the lower concentrations (0.7, 0.6 µL/mL) (Figure 2). The IC_50_ values obtained were 0.6 µL/mL for U87, U118, and U373 cells, 0.7 µL/mL for A172 cells, and 0.8 µL/mL for H4 cells. We believe that these results are due to the content of camphor (39%) and 1,8-cineole (10.9%) already described for their therapeutic bioactivities. 

There are other studies that sustain this structure–anticancer activity relationship. For example, the antiproliferative property of *Salvia officinalis* EO was tested against three human colon cancer cell lines (Caco-2, HT-29, and HCT-116), and the results revealed that this EO has an in vitro antiproliferative effect due to its composition rich in 1,8-cineole, camphor, and α-thujone [60]. Another example is the work of Lima et al. [61] that evaluated the anticancer activity of *Ocimum kilimandscharicum* Gürke EO on ten human cancer cell lines (U251, UACC-62, MCF-7, NCI-ADR/RES, 786-0, NCI-H460, PC-3, OVCAR-03, k-562, and HT-29). This in vitro cytotoxic screening demonstrated high selectivity and potent anticancer activity against the cell lines studied, particularly the OVCAR-3 cells. Based on its EO composition and the therapeutic activity observed, authors assumed that the presence of camphor and 1,8-cineole is responsible for the bioactivity verified [61]. Therefore, these studies corroborate our results with the chemical composition of *L. pedunculata* EO being rich in camphor and 1,8-cineole and, in turn, linked to a great cytotoxic/antiproliferative activity.

Treatment with *M. cervina* EO induced a strong antiproliferative/cytotoxic effect against all five GB cell lines (A172, U87, H4, U373, and U118 cells), without significantly affecting the viability of non-tumor cells (HEK-293 cell line) (Figure 3). Particularly, at the 24 h time-point (after treatment with the *M. cervina* EO), tumor cells showed a significant (**** *p* < 0.0001) decrease in their metabolic activity for the higher concentrations tested (1.0, 0.9, 0.8 µL/mL) (Figure 3). Although, after 48 h treatment, an increase of the viability in tumoral cells was observed. The amelioration detected in the metabolic activity of GB cells may be related to the chemical nature of EOs and their physicochemical properties, such as volatility. Bearing this in mind, the results obtained for this time point are in accordance with what was expected. At 72 h post-incubation, the metabolic activity of A172, U373, and U118 cells suffer a slight decrease with the lower concentrations (0.7, 0.6 µL/mL), when compared to the previous time-point. For both U87 and H4 cells, even though a reduction in metabolic activity was observed, cell viability increased with some exceptions (0.6 and 1.0 µL/mL) (Figure 3). These data indicate that although *M. cervina* EO exhibited a pronounced antiproliferative/cytotoxic effect against GB cells, it did not affect non-tumor cells, which reveals a potential tumor-specific mechanism. Moreover, this EO’s half-inhibitory concentration (IC_50_) in tumor cells (A172, H4, U373, and U118 cells-0.7 µL/mL; U87 cells-0.8 µL/mL) was not cytotoxic for HEK-293 cells. These results were expected, since this EO has compounds with already described anticancer effects, such as pulegone, but with relatively low cytotoxicity in non-tumor cells. In a study where the antitumor activity of several EOs was evaluated against different cancer cells (MCF-7, HeLa, Jurkat, HT-29, and T24 cell lines), and one non-tumor cell line (HEK-293 cells), authors verified that EO from *Origanum vulgare* L. (*O. vulgare*) had a strong cytotoxic effect against cancer cells, but not against non-tumor cells. They concluded that this tumor-specific effect could be attributed to the high content of pulegone (77.45%) present in the *O. vulgare* EO [62]. This supports results obtained in this study (Figure 3). Rahimifard et al. [63] also explored the cytotoxic effect of *Mentha pulegium* L. (*M. pulegium*), a species with its EO enriched in pulegone, in cancer cells. Once more, it was stated that this EO induced a pronounced decrease in the viability of cancer cells [63]. In another study, the presence of a high percentage (58.5%) of pulegone in *Micromeria fruticosa* (L.) Druce subsp. *serpyllifolia* EO was also associated to improved antitumor activity [64].

At last, the cytotoxic profile of *T. mastichina* EO was also assessed against GB cell lines and non-tumor cells (Figure 4). As already observed for the previously analyzed EOs, right after 24 h treatment with *T. mastichina* EO, a significant (* *p* < 0.05; ** *p* < 0.01; **** *p* < 0.0001) impact on cell viability was observed, with a decrease in the metabolic activity of tumor cells (Figure 4). Although, it is worth mentioning that this EO, like that verified for *M. cervina*, did not significantly affect the viability of non-tumor cells (Figure 4). Once more, this tumor-specific profile is very interesting and worth exploring. Although, after 48 h and 72 h treatment, tumor cells seem to recover their proliferative activity (Figure 4). Furthermore, the IC_50_ values were 0.9 µL/mL for U87, H4, U118, and A172 cells and 0.8 µL/mL for U373 cells. Regarding this, the results obtained for *T. mastichina* EO are in accordance with data available in the literature. For example, Loizzo et al. [65] evaluated the cytotoxic activity of EOs from different plant species in amelanotic melanoma (C32), renal cell adenocarcinoma (ACHN), hormone-dependent prostate carcinoma (LNCaP) and breast cancer (MCF-7) cell lines. Out of those plants, both *Laurus nobilis* L. (*L. nobilis*) and *Salvia officinalis* L. (*S. officinalis*) EOs were rich in 1,8-cineole (35.15% and 43.62%, respectively). Interestingly, EO obtained from *L. nobilis* fruit was the most active on the amelanotic melanoma cells and renal adenocarcinoma cells, while less activity was found when the EO from *L. nobilis* leaf was applied to both cell cultures. This difference in the cytotoxicity of the different EOs might be attributed to the different compositions of EOs obtained from different plant organs. Furthermore, *S. officinalis* EO exhibited its lowest IC_50_ values in renal cell adenocarcinoma. However, when authors tested the commercially available sample of the 1,8-cineole compound on several cancer cell lines, they surprisingly discovered that this monoterpene was inactive against the cell lines under study, which may hinder the importance of a synergistic activity of different compounds present in the EOs [65]. 

The referred association between the different major monoterpenes and the antitumor activity attributed to these EOs was considered in the development of this work. Therefore, after the treatment applied to GB cells, either with *L. pedunculata* EO, *M. cervina* EO or *T. mastichina* EO, a therapeutic effect was observed by the significant decrease in cell viability and proliferative capability of cancer cells, this effect being more significative with *L. pedunculata* and *M. cervina* EOs. Therefore, to better understand the mechanism of these EOs, further analysis to assess the cell death mechanism is of great interest.

### 3.4. Assessment of Cell Death Profile and Cell Cycle Progression of Essential Oils

According to the cell viability analysis, from the three species under study, *L. pedunculata* and *M. cervina* EOs presented higher cytotoxic/antiproliferative effects against GB cells, probably due to their chemical composition. Thereby, the next step was to further assess the mechanism by which these EOs induced the decrease of cell viability on a gold standard GB cell model (U87 cell line). Based on that, a cell death profile was made using AV/PI assay, which enables us to distinguish live cells from cells in apoptosis and from cells in necrosis. 

AV-binding assay is one of the most used experiments to detect apoptosis, via the activation of caspases. In apoptotic stages, caspases cleave the lipid flippases in the inner leaflet of the plasma membrane, which exposes the phosphatidylserine (PS) to the outer leaflet layer of the cell membrane. Thus, at this stage, the PS is recognized by the AV (stoichiometric binding reaction), an indicative process of early apoptosis [45,66]. On the other hand, in late apoptosis and/or necrosis, the integrity of the plasma and nuclear membranes is greatly decreased, making it more permeable and therefore allowing for the passage of the PI fluorochrome. Once the PI passes through the membrane, it binds to the DNA and stains the nucleus. It is important to note, however, that this process only happens in cells whose membranes have become permeable (non-viable cells), and it cannot happen in live or in early apoptotic cells due to the presence of an intact plasma membrane [45,67,68]. Thus, by performing a double staining assay based on these two methods, it is possible to ascertain if cells are viable, apoptotic, or necrotic through alterations in plasma membrane integrity and permeability, since the transport of PS to the outer cell membrane allows for the penetration of the PI fluorochrome into the cell in a unidirectional manner.

The analysis of the results obtained showed that cells treated with both concentrations, 0.7 and 0.8 µL/mL, of either *L. pedunculata* or *M. cervina* EOs promoted a significant decrease in the % of live cells (Figure 5), which supports our previous results (Figure 2 and Figure 3). Moreover, treatment with both EOs led to a substantial increase of cells in late apoptosis/necrosis, which was accompanied by a significant increase in the % of cells in early apoptosis (Figure 5). However, 0.7 µL/mL of *L. pedunculata* EO was the exception to this pattern, with only a significant increase of cells in late apoptosis/necrosis (Figure 5). Once more, these data are in accordance with literature already reported for EOs with a similar chemical composition. As an example, Duymus et al. [69] reported the EO from *Vitex agnus-castus* L. fruit, enriched in 1,8-cineole, showed high apoptotic activity against both A549 (human lung adenocarcinoma cells) and MCF-7 cells [69]. Another work stated that treatment with *Ziziphora tenuior* L. EO (with a high content in pulegone) on colorectal cancer epithelial cells (HT-29) revealed a time-dependent apoptosis mechanism [70]. It is worth mentioning that the reason for these apoptotic events in these studies seems to be due to the rich content in camphor and pulegone from *L. pedunculata* and *M. cervina*, respectively. Considering these results (Figure 5), the most probable mechanism for both EOs is via inducing cell death by apoptosis due to the substantial increase of cells in late and early apoptosis. Nevertheless, an analysis of the impact of these EOs on cell cycle could help to sustain our preliminary conclusions.

The impact of *L. pedunculata* and *M. cervina* EOs on cell cycle progression was evaluated using PI/RNase assay by flow cytometry. As described in the previous assay, U87 cells were treated with 0.7 and 0.8 µL/mL of either *L. pedunculata* or *M. cervina* EOs. Vehicle (DMSO) and untreated cells were considered the control groups (Figure 6).

When treating the U87 cells with *M. cervina* EO, results obtained are very similar to the ones described by the effect of *L. pedunculata* EO. Treatment with 0.8 µL/mL of both EOs promoted an increasing percentage of cells in subG0/G1 (Figure 6), meaning an increase of cells in early apoptosis. This could explain why the increasing % of cells observed in late apoptosis/necrosis (Figure 5) undergo cell death by apoptosis. Furthermore, it was also observed a block of cells in S and G2/M phases for this higher concentration. Interestingly, for the lower concentration (0.7 µL/mL), this profile was not verified. In fact, it seems that with this concentration both EOs have the main impact on block G2/M phase (Figure 6).

Thus, these data allowed to conclude that both EOs have a dose-dependent effect, and the most likely mechanism is inducing cell death by apoptosis through DNA damage and downregulation of the cyclin-dependent kinase1 (cdk1)/cyclin B1 complex with an enhancement of p21 expression [11,71].

## 4. Conclusions

In this study, important insights on *L. pedunculata*, *M. cervina* and *T. mastichina* EOs were provided, thus paving the way for the design of innovative and alternative antitumor drugs. Overall, from the three plants under study, the EOs from both *L. pedunculata* and *M. cervina* stand out with stronger cytotoxic activity against GB cells, a bioactive profile probably associated with their rich composition in pulegone and camphor, respectively. Furthermore, both EOs presented a dose-dependent antitumor effect, inducing cell death by apoptosis. Despite achieved results, further studies will be performed to better understand the antitumor mechanism of action. It is however clear that this is a very interesting finding focusing on natural compounds’ contribution in drug discovery for glioblastoma treatment.

## Figures and Tables

**Figure 1 pharmaceutics-15-00341-f001:**
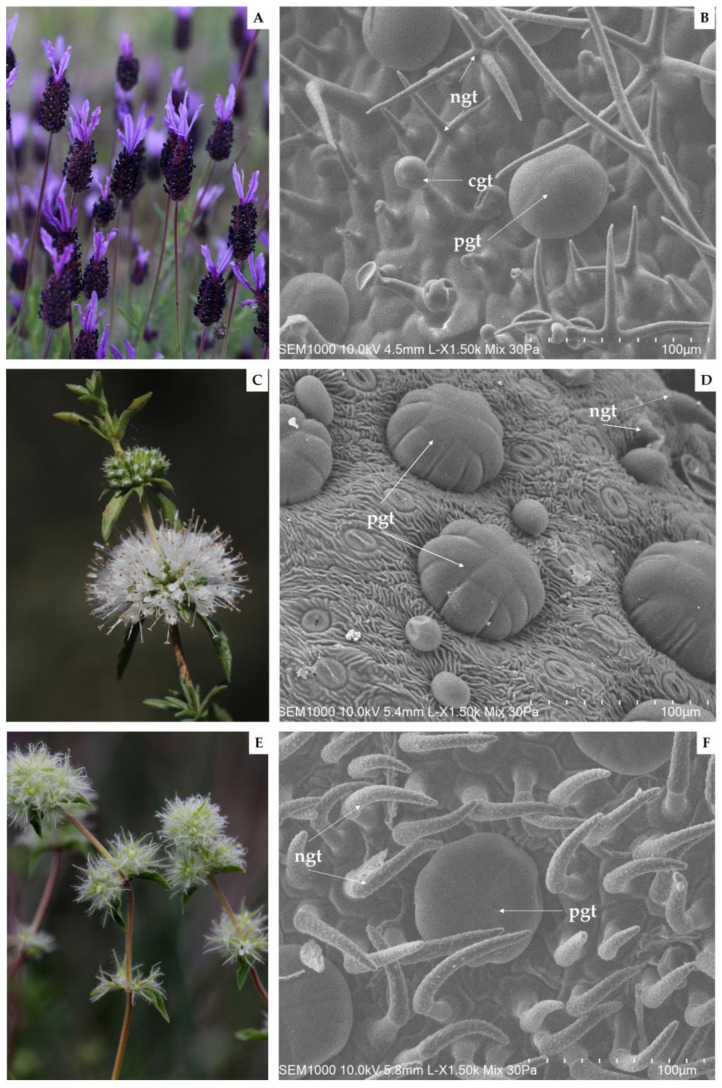
Photographic records of Côa Valley Lamiaceae species in flowering stage and respective scanning electron micrographs captured from their abaxial leaf surfaces, as follows: *L. pedunculata* (**A**,**B**), *M. cervina* (**C**,**D**), and *T. mastichina* (**E**,**F**). Note: *cgt*—capitate glandular trichomes; *ngt*—non-glandular trichomes; and *pgt*—peltate glandular trichomes.

**Figure 2 pharmaceutics-15-00341-f002:**
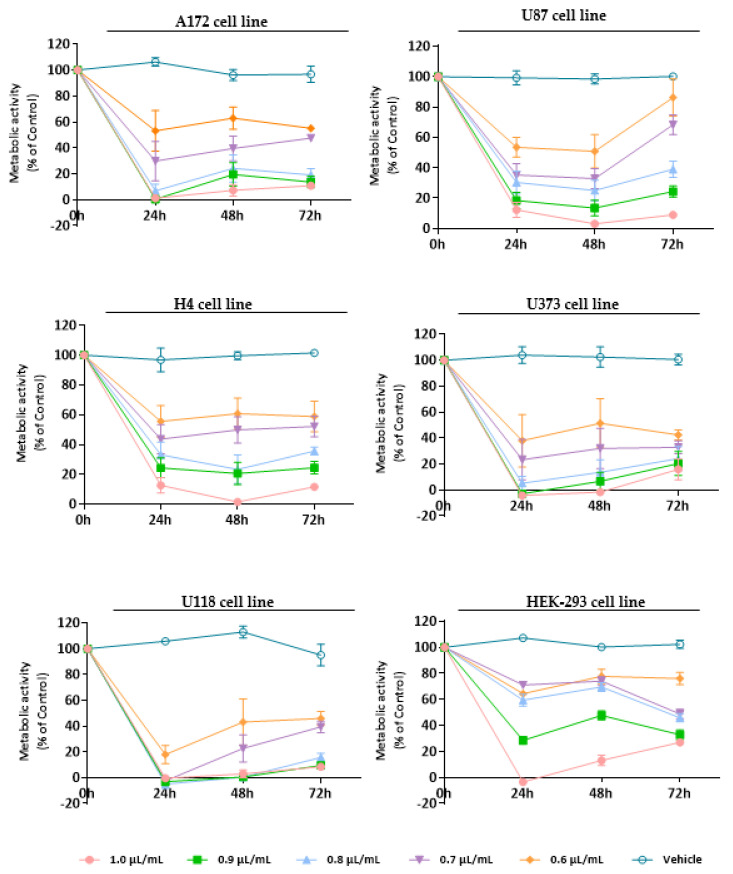
Cytotoxic effect of *L. pedunculata* EO against GB cells (A172, U87, H4, U373, and U118 cell lines) and non-tumoral cells (HEK-293 cell line). Cells were treated either with *L. pedunculata* EO (1.0, 0.9, 0.8, 0.7, and 0.6 µL/mL) or with the vehicle control (DMSO) for 24, 48, and 72 h. Cell viability (metabolic activity) was expressed as percentage of control (untreated cells). Results were presented as mean ± SD.

**Figure 3 pharmaceutics-15-00341-f003:**
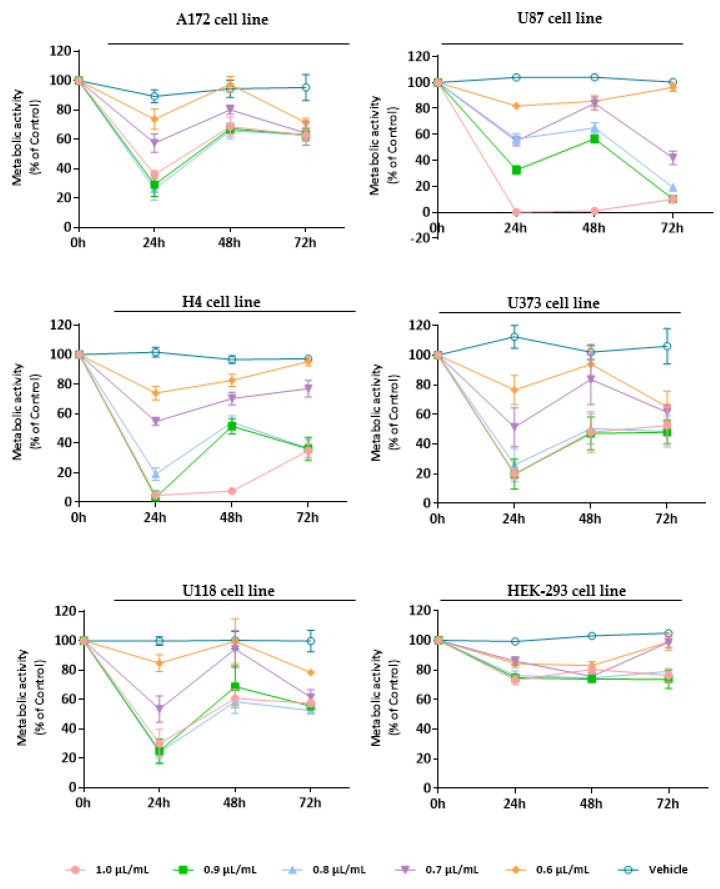
Cytotoxic effect of *M. cervina* EO against GB cells (A172, U87, H4, U373, and U118 cell lines) and non-tumoral cells (HEK-293 cell line). Cells were treated either with *M. cervina* EO (1.0, 0.9, 0.8, 0.7, and 0.6 µL/mL) or with the vehicle control (DMSO) for 24, 48, and 72 h. Cell viability (metabolic activity) was expressed as percentage of control (untreated cells). Results were presented as mean ± SD.

**Figure 4 pharmaceutics-15-00341-f004:**
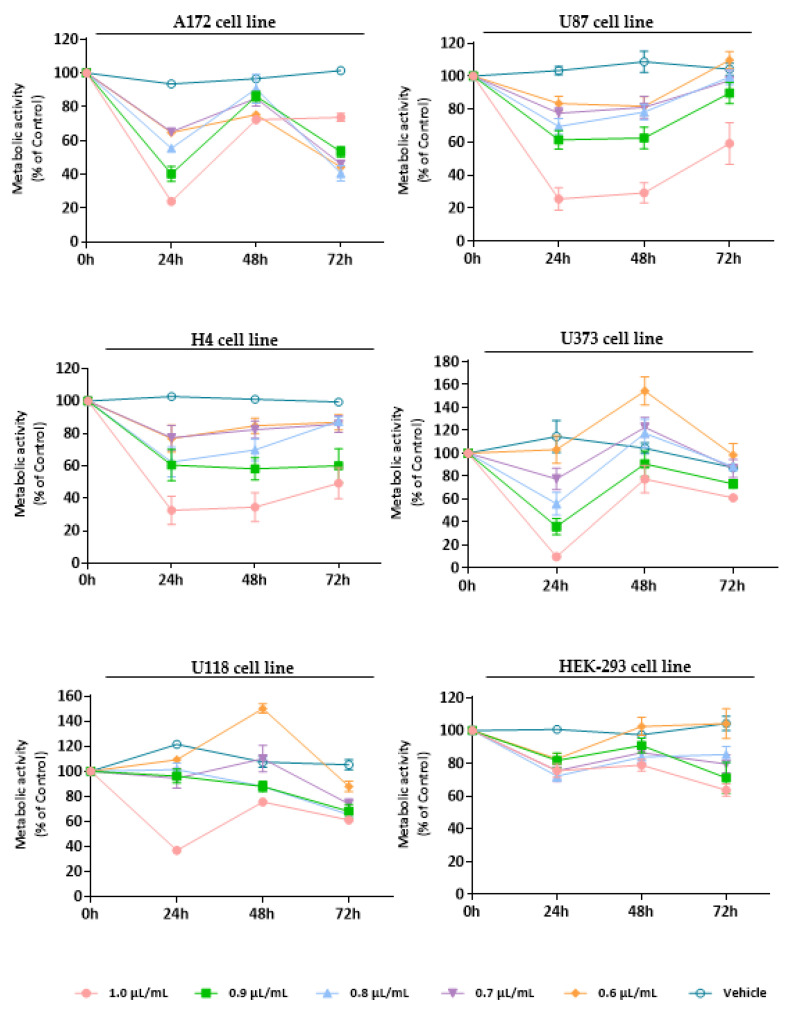
Cytotoxic effect of *T. mastichina* EO against GB cells (A172, U87, H4, U373, and U118 cell lines) and non-tumoral cells (HEK-293 cell line). Cells were treated either with *T. mastichina* EO (1.0, 0.9, 0.8, 0.7, and 0.6 µL/mL) or with the vehicle control (DMSO) for 24, 48, and 72 h. Cell viability (metabolic activity) was expressed as percentage of control (untreated cells). Results were presented as mean ± SD.

**Figure 5 pharmaceutics-15-00341-f005:**
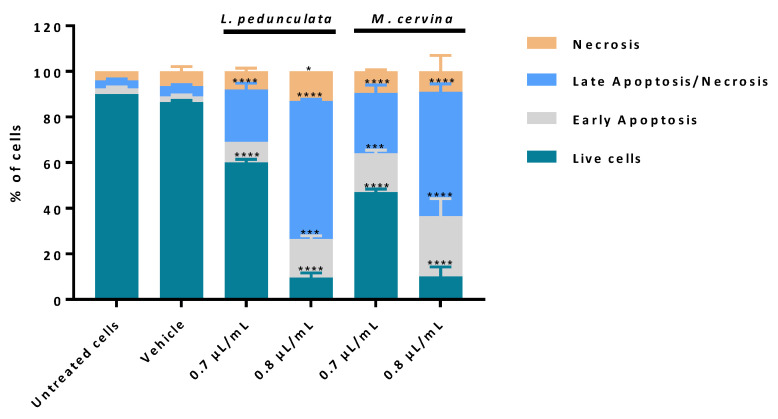
Assessment of cell death profile in U87 cell line by flow cytometry. Cells were treated either with *L. pedunculata* EO and *M. cervina* EO or with the vehicle control (DMSO), and further incubated for 24 h. After incubation, cells were co-stained with Annexin V and PI and the percentage of cells in apoptosis and non-apoptotic cells was determined. Asterisks (* *p* < 0.05, *** *p* < 0.001, and **** *p* < 0.0001) represent the values that significantly differ from the control (untreated cells). Data are expressed as mean ± SD.

**Figure 6 pharmaceutics-15-00341-f006:**
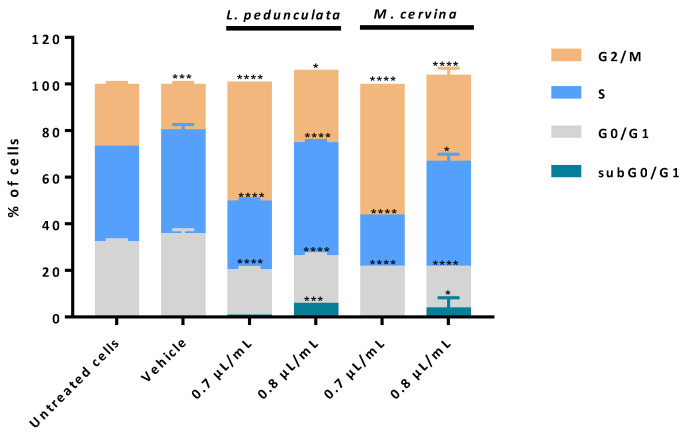
Analysis of cell cycle progression in U87 cell line. Cells were treated either with *L. pedunculata* EO and *M. cervina* EO or with the vehicle control (DMSO), and further incubated for 24 h. Cells were stained with PI/RNase solution and cell cycle analysis was assessed by flow cytometry. Proportion of cells in subG0/G1, G0/G1, S and G2/M cell cycle phases was expressed in percentage of total cell population. Asterisks (* *p* < 0.05, *** *p* < 0.001, and **** *p* < 0.0001) represent the values that significantly differ from the control (untreated cells). Data are expressed as mean ± SD.

**Table 1 pharmaceutics-15-00341-t001:** Chemical composition of *L. pedunculata*, *M. cervina*, and *T. mastichina* OEs from Côa Valley, Northeastern Portugal.

Compound	RT	LRI ^a^	LRI ^b^	LP ^c^	MC ^d^	TM ^e^
Tricyclene	13.29	914	921	0.39	-	0.06
α-Thujene	13.61	920	924	-	-	0.18
α-Pinene	13.91	925	932	6.9	0.66	4.1
Camphene	14.65	940	946	4.00	-	1.4
2,4(10)-Thujadiene	14.97	946	953	0.357	-	-
Sabinene	15.99	965	969	0.022	0.45	3.2
β-Pinene	16.12	968	974	0.107	1.139	5.2
2,3-Dehydro-1,8-cineole	16.88	983	988	-	-	0.34
β-Myrcene	16.98	984	988	-	0.2	1.1
Δ^3^-Carene	17.89	1002	1008	1.21	-	-
4-Carene	18.24	1009	1018 *	0.048	0.85	0.27
*p*-Cymene	18.52	1014	1020	0.28	-	-
*o*-Cymene	18.66	1017	1022	0.53	0.36	0.61
Limonene	18.8	1019	1025	0.53	1.46	-
1,8-Cineole	19.03	1024	1026	10.9	-	57.00
*cis*-β-Ocimene	19.41	1031	1032	0.18	0.161	0.18
*trans*-β-Ocimene	19.93	1041	1044	-	-	1.7
γ-Terpinene	20.44	1051	1053	0.092	1.6	0.61
*cis*-Sabinene hydrate	20.86	1059	1065	-	3.4	0.54
*cis*-Linalool oxide	21.14	1065	1067	0.68	-	0.68
Camphenilone	21.65	1074	1078	0.45	-	0.45
Fenchone	21.9	1079	1083	5.36	-	5.36
*p*-Mentha-2,4(8)-diene	21.97	1081	1085	-	0.4	0.41
*trans*-Sabinene hydrate	22.45	1090	1096 *	-	0.28	0.28
Linalool	22.56	1091	1095	1.9	0.18	1.29
Fenchol	23.25	1105	1114	0.072	-	0.072
*cis*-*p*-Menth-2-en-1-ol	23.62	1113	1120 *	-	0.23	0.144
α-Campholenal-(+)2-pinen-7-one	23.85	1118	1122 + 1124	0.62	-	0.62
*trans*-Pinocarveol	24.50	1131	1135	0.19	-	0.17
*trans*-*p*-Menth-2-en-1-ol	24.55	1132	1139 *	-	0.24	-
Camphor	24.73	1135	1141	39.00	-	1.2
Menthone	25.24	1146	1148	-	0.12	-
Pinocarvone	25.66	1154	1160	0.12	-	0.102
Menthofuran	25.74	1156	1159	-	0.19	-
*endo*-Borneol	25.88	1159	1165	2.4	-	4.8
Isopulegone	26.33	1168	-	-	2.55	-
Terpinen-4-ol	26.41	1169	1174	0.66	5.7	1.8
*m*-Cymen-8-ol	26.64	1174	1176	0.6	-	-
*p*-Cymen-8-ol	26.79	1177	1179	1.1	-	-
α-Terpineol	27.07	1182	1186	0.69	0.5	7.2
Myrtenal	27.32	1188	1195	0.37	-	0.43
Verbenone	27.93	1200	1204	1.2	-	-
Pulegone	29.45	1230	1233	-	60.00	-
Linalool acetate	30.17	1248	1254	0.25	-	-
Bornyl acetate	31.59	1278	1284	5.9	-	0.07
Lavandulyl acetate	31.8	1282	1288	0.28	-	0.28
α-Terpinyl acetate	34.45	1341	1346	-	-	0.19
β-Elemene	36.35	1385	1389	-	0.18	-
β-Caryophyllene	37.56	1412	1417	-	1.2	0.5
Alloaromadendrene	39.31	1455	1458	-	0.218	0.11
Cadina-3,5-diene	39.38	1456	-	0.07	-	-
Germacrene D	40.13	1474	1477	0.22	12.00	-
β-Selinene	40.36	1480	1489	0.05	-	-
1,11-Oxidocalamenene	40.51	1483	1490 *	0.42	-	-
Bicyclogermacrene	40.77	1490	1500	-	1.3	0.26
γ-Cadinene	41.46	1507	1513	0.12	-	-
δ-Cadinene	41.81	1516	1522	0.38	0.6	-
Elemol	42.84	1541	1548	-	-	0.9
Globulol	44.57	1585	1590	-	2.00	1.5
Ledol	45.02	1597	1602	-	-	0.22
1,10-Di-*epi*-cubenol	45.43	1609	1618	0.37	-	-
γ-Eudesmol	46.07	1625	1630	-	0.3	0.24
Monoterpene hydrocarbons				14.7	10.9	19,00
Oxygen-containing monoterpenes				73.00	70.00	74.7
Sesquiterpene hydrocarbons				0.8	15.00	0.9
Oxygen-containing sesquiterpenes				7.3	2.7	2.8
Total identified				95.80	98.60	97.40

^a^ LRI: linear retention index determined on a SH-RXi-5 ms fused silica column (Shimadzu) relative to a series of n-alkanes (C8–C40); ^b^ LRI: linear retention index reported in the literature [57] or * retrieved from NIST database [58]; ^c,d,e^ Abbreviations: LP-*Lavandula pedunculata*, MC-*Mentha cervina,* and TM-*Thymus mastichina.*

## Data Availability

The data presented in this study are available on request from the corresponding author.

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
