# Peer review of "Essential Oils from Côa Valley Lamiaceae Species: Cytotoxicity and Antiproliferative Effect on Glioblastoma Cells"

_pharmaceutics, 2023, doi:10.3390/pharmaceutics15020341_

Round 1

Reviewer 1 Report

In this paper, the authors designed the L. pedunculata, T. mastichina and M. cervine have traditional uses, these aromatic plants are still undervalued and literature regarding their bioactivity, especially anticancer, is scarce. In this study, the morphology of secretory structures was assessed by scanning electron microscopy (SEM) and the composition of essential oils (EOs) was characterized by gas chromatography-mass spectrometry (GC-MS). The study proceeded with cytotoxic evaluation of EOs in tumor and non-tumor cells with cell death mechanism explored in glioblastoma (GB) cells. L. pedunculata’s EO presented the most pronounced cytotoxic/antiproliferative activity against tumor cells, with moderate cytotoxicity against non-tumor cells. Meanwhile, M. cervina’s EO exhibited a slightly lower cytotoxic effect against tumor cells and did not affect the viability of non-tumor cells. These two EOs lead to cell death by inducing apoptosis in a dose-dependent manner. The present study suggests that L. pedunculata and M. cervina’s EOs have a strong cytotoxic and antiproliferative potential to be further studied as efficient antitumor agents. However, numerous mistakes would misunderstand and confuse readers. Several typo-error should be re-checked and revised. Since, I recommended accepted this paper after minor revision. Some corrections and suggestions were listed below:

(1)   In Table 1, all the decimal point should be used correctly. For example, the Camphene in LP should be revised from 4,00 to 4.00.

(2)   In Table 1, all the abbreviation should be marked with superscript and described in footnote.

(3)   In Table 1, the orientation of substituent of compounds should be written in ‘italics’. For example, the p-Cymene, o-Cymene, m-Cymen-8-ol, and p-Cymen-8-ol should be p-Cymene, o-Cymene, m-Cymen-8-ol, and p-Cymen-8-ol.

(4)   In Figure 2–4, the size of X and Y axis should be amplified.

(5)   In the line 521, the repeated [] should be revised, for example, [[44,73]] should be revised to [44,73].

(6)   The name of each cell line should be consistent. For example, HEK293 or HEK-293.

(7)   The space should be removed in Line 404.

Author Response

Response to reviewers’ comments (author responses are noted in italic)

We thank the reviewers for their time reading our manuscript, and for their valuable comments and suggestions. We are grateful for these comments, and the article suffer an improvement as a result. In the response to the comments below, we hope to have answered to all queries noted, as part of the revision process for the manuscript. We have addressed them either in the manuscript or in the rebuttal letter below, as relevant.

Comments from Reviewer 1

We thank the referee for the valuable comments and suggestions, and we have responded to each comment below.

In Table 1, all the decimal point should be used correctly. For example, the Camphene in LP should be revised from 4,00 to 4.00.

Thank you for your correction. Table 1 was properly revised, being all the commons changed to points according to the reviewer’s suggestion.

In Table 1, all the abbreviation should be marked with superscript and described in footnote.

Thank you for your suggestion. All the abbreviations in Table 1 were marked with superscript letters (a, b, c, d, e) and described in footnote as suggested.

In Table 1, the orientation of substituent of compounds should be written in ‘italics’. For example, the p-Cymene, o-Cymene, m-Cymen-8-ol, and p-Cymen-8-ol should be p-Cymene, o-Cymene, m-Cymen-8-ol, and p-Cymen-8-ol.

Thank you for your comment. We addressed this mistake and change the orientation of substituent of compounds to italic as properly suggested.

In Figure 2–4, the size of X and Y axis should be amplified.

We increased the size of X and Y axis in Figure 2-4 as suggested.

In the line 521, the repeated [] should be revised, for example, [[44,73]] should be revised to [44,73].

We addressed this mistake and the repeated [] were deleted as suggested.

The name of each cell line should be consistent. For example, HEK293 or HEK-293.

Thank you for your comment. To maintain the consistency in the manuscript, the name of the cell line was corrected for HEK-293 as suggested.

The space should be removed in Line 404.

The space was removed as suggested.

Reviewer 2 Report

The study entitled Essential oils from Côa Valley Lamiaceae species: cytotoxicity 2 and antiproliferative effect on glioblastoma cells presents scientific relevance and contributes to the knowledge about plants essential oils and their cytotoxic tumor cell properties. The authors explored important preserved regional flora of their country, contributing to ethnobotanical aspects. It was performed a detailed screening about composts of EO contained in three plants and their electron microscopy. The tumor cells study were glioblastomas that is an aggressive type of cancer, with difficulty to treatment. The study delineated since cytotoxic/antiproliferative effects against, glioblastoma cells, as non-tumor cells. Besides, it was evidenced the mechanism of the cytotoxic effect against tumor. Finally, the results are promising and the discussion is sufficiently substantiated.

It is listed below, minor considerations in order to improve the manuscript quality.

1. Standardize the information about each author - or everybody in English or in Portuguese.

Abstract

2.The authors did not mention anything in the results about Thymus mastichina EO.

Material em methods

3. Which were the part of the plants used? Whole plant?

4. Statistical analysis – why did it perform 2-way ANOVA? What was the second variable to compared?

Results

5. It is important to highlight also the second and thirth major components of T. mastichina’s EO - α-Terpineol and Fenchone, respectively.

6. The sequence of described results should be according to the table 1, L. pedunculata (LP), M. cervina (MC), and T. mastichina (TM), respectively and not L. pedunculata (LP), M. mastichina (TM), and T. cervina (MC). The same could be applied to figure 2, figure 3 and figure 4, first L. pedunculata (LP)-figure 2, second M. cervina (MC)-figure 3, and third T. mastichina (TM)-figure 4.

7. In figure 2 the authors described that there are not cytotoxic effects of M. cervina to non-tumor cells (HEK 293). However, the cell metabolic activity was 70% of the control vehicle 24 h after the incubation. The same occurred to M. mastichina.

8. Figure 5 – the number of the lines insert in the figure harmful the visualization.

Author Response

Response to reviewers’ comments (author responses are noted in italic)

We thank the reviewers for their time reading our manuscript, and for their valuable comments and suggestions. We are grateful for these comments, and the article suffer an improvement as a result. In the response to the comments below, we hope to have answered to all queries noted, as part of the revision process for the manuscript. We have addressed them either in the manuscript or in the rebuttal letter below, as relevant.

Comments from Reviewer 2

We thank the referee for the valuable comments and suggestions, and we have answered to each comment below.

Standardize the information about each author - or everybody in English or in Portuguese.

Thank you for your comment. The information regarding each author affiliation was changed to English to maintain consistency as suggested.

Abstract

The authors did not mention anything in the results about Thymus mastichina EO.

Thank you for your comment. The following sentence “Meanwhile, T. mastichina EO did not induce a strong cytotoxic effect against GB cells.” was added to the abstract section as properly suggested.

Material and methods

Which were the part of the plants used? Whole plant?

Thank you for raising a valuable question. The essential oils used in this study were extracted from the aerial parts of each plant and these plants were collected during the flowering stage.

Statistical analysis – why did it perform 2-way ANOVA? What was the second variable to compared?

Thank you for your comment. The statistical analysis performed was 1-way ANOVA, instead of 2-way ANOVA, we acknowledge this mistake, and the proper corrections were made in the section “2.9. Statistical Analysis”.

Results

It is important to highlight also the second and third major components of T. mastichina’s EO - α-Terpineol and Fenchone, respectively.

Thank you for your suggestion. The following sentence was added to the section 3.1 Essential oil composition “This EO has 1,8-cineole as main compound with 57%, followed by α-terpineol (7.2%) and fenchone (5.36%).” to highlight these compounds as suggested.

The sequence of described results should be according to the table 1, L. pedunculata (LP), M. cervina (MC), and T. mastichina (TM), respectively and not L. pedunculata (LP), M. mastichina (TM), and T. cervina (MC). The same could be applied to figure 2, figure 3 and figure 4, first L. pedunculata (LP)-figure 2, second M. cervina (MC)-figure 3, and third T. mastichina (TM)-figure 4.

We addressed this mistake, and the sequence of the described results was changed as suggested.

In figure 2 the authors described that there are not cytotoxic effects of M. cervina to non-tumor cells (HEK 293). However, the cell metabolic activity was 70% of the control vehicle 24 h after the incubation. The same occurred to M. mastichina.

Thank you for your comment. In the discussion of these results, we described that M. cervina and T. mastichina EOs did not significantly affect the viability of non-tumor cells. This because, we considered that the impact of these EOs in the viability of non-tumor cells were not significant relatively to control since the cell viability was maintained constant and did not drop below 70%. Although, we acknowledge that cell viability was slightly affected, but this effect was not significant.

Figure 5 – the number of the lines insert in the figure harmful the visualization.

Thank you for your comment. The graph presented in Figure 5 is a stacked bar graph. In this type of graph, each coloured region of a bar corresponds to a data group, being this data group identified on the right side of the graph.

Reviewer 3 Report

I have read the manuscript titled “Essential oils from Côa Valley Lamiaceae species: cytotoxicity and antiproliferative effect on glioblastoma cells”. The manuscript is devoted to the study of the cytotoxic and antiproliferative effects of Lavandula pedunculata, Mentha cervina and Thymus mastichina essential oils on glioblastoma cells. The authors also studied the chemical composition of essential oils of these species and the morphology of secretory structures.

The manuscript is interesting and well-written. However, I have some minor comments:

1.      Include the information of cytotoxic effect of Lavandula pedunculata essential oil against non-tumoral cells (HEK-293 cell line) in Discussion.

2.      I recommend that authors indicate the names of the studied plant species in the title or in keywords. This will make it easier to find information later.

3.      Please indicate the geographical coordinates of the area where the raw materials were collected.

4.      Table 1 should be brought to uniformity. After the comma there were 2 or 3 signs (usually 1), instead of a dot there was a comma. Also bring to uniformity the spelling of chemical compounds (“gamma-terpinene”, “-terpinyl acetate” etc.).

I wish the authors success!

Author Response

Response to reviewers’ comments (author responses are noted in italic)

We thank the reviewers for their time reading our manuscript, and for their valuable comments and suggestions. We are grateful for these comments, and the article suffer an improvement as a result. In the response to the comments below, we hope to have answered to all queries noted, as part of the revision process for the manuscript. We have addressed them either in the manuscript or in the rebuttal letter below, as relevant.

Comments from Reviewer 3

We thank the referee for the valuable comments and suggestions, and we have addressed each comment below.

Include the information of cytotoxic effect of Lavandula pedunculata essential oil against non-tumoral cells (HEK-293 cell line) in Discussion.

Thank you for your suggestion. Information regarding the cytotoxic effect of L. pedunculata EO against HEK-293 cell line was provided in the section “3.3. Evaluation of the cytotoxic Effect of Essential Oils” (line 285 to 287) as suggested.

I recommend that authors indicate the names of the studied plant species in the title or in keywords. This will make it easier to find information later.

Thank you for your suggestion. We removed “Lamiaceae” from the keywords and added the name of the three studied plant species as suggested.

Please indicate the geographical coordinates of the area where the raw materials were collected.

We provided detailed information regarding the areas where the plants were collected in the section “2.1. Plant Material” as suggested.

Table 1 should be brought to uniformity. After the comma there were 2 or 3 signs (usually 1), instead of a dot there was a comma. Also bring to uniformity the spelling of chemical compounds (“gamma-terpinene”, “-terpinyl acetate” etc.).

Thank you for your comment. We carefully reviewed Table 1 and addressed these mistakes by formatting Table 1 to maintain consistency and uniformity as suggested.
